

**Title:**
**Changes in soil carbon and nutrients following six years of litter removal and addition in a tropical**
**semi-evergreen rain forest.**
**Authors**
**Edmund Vincent John Tanner[1,2], Merlin William Alfred Sheldrake[1], and Benjamin L. Turner[2]**
**[1]Department of Plant Sciences, University of Cambridge, Downing St, Cambridge CB2 3EA, UK.**
**[2]Smithsonian Tropical Research Institute, Apartado 0843-03092, Balboa, Ancon, Republic of**
**Panama.**
*Correspondence to*: E. V. J. Tanner (evt1@cam.ac.uk)
**Abstract**
Increasing atmospheric $CO_2$ and temperature may increase forest productivity, including litterfall,
but the consequences for soil organic matter remain poorly understood. To address this, we
measured soil carbon and nutrient concentrations at nine depths to 2 m after six years of continuous
litter removal and litter addition in a semi-evergreen rain forest in Panama. Soils in litter addition
plots, compared to litter removal plots, had higher pH and contained greater concentrations of KCl-
extractable nitrate (both to 30 cm); Mehlich-III extractable phosphorus and total carbon (both to 20
cm); total nitrogen (to 15 cm); Mehlich-III calcium (to 10 cm); Mehlich-III magnesium and lower bulk
density (both to 5 cm). In contrast, litter manipulation did not affect ammonium, manganese,
potassium or zinc, and soils deeper than 30 cm did not differ for any nutrient. Comparison with
previous analyses in the experiment indicates that overall the effect of litter manipulation on
nutrient concentrations and the depth to which the effects are significant are increasing with time.
To allow for changes in bulk density in calculation of changes in carbon stocks, we standardized total
carbon and nitrogen on the basis of a constant mineral mass. For 200 kg m$^{-2}$ of mineral soil
(approximately the upper 20 cm of the profile) about 0.5 kg C m$^{-2}$ was 'missing' from the litter
removal plots, with a similar amount accumulated in the litter addition plots. There was an
additional 0.4 kg C m$^{-2}$ extra in the litter standing crop of the litter addition plots compared to the
control. This increase in carbon in surface soil and the litter standing crop can be interpreted as a
potential partial mitigation of the effects of increasing $CO_2$ concentrations in the atmosphere.

**1  Introduction**
Tropical forests and their soils are an important part of the global carbon (C) cycle, because they
contain 692 Pg C (two thirds in evergreen and one third in deciduous forests), equivalent to 66 % of
the C in atmospheric $CO_2$ (Jobbagy and Jackson 2000). Carbon in tropical forest soils is dynamic;
Schwendenmann and Pendall (2008) reported a turnover time of 15 years for the 'slow' pool of soil C
(38 % of the total soil C; 61% of total soil C was 'passive' with a turnover time of the order of a
thousand years) in the top 10 cm of soil in semi-evergreen rain forest on Barro Colorado Island,
Panama. Turner et al. (2015) reported an approximate 25% increase in soil C from one dry season to
the next wet season in the top 10 cm of soil on the Gigante Peninsula in Barro Colorado Nature
Monument, Panama; a site near where the current litter manipulation experiment was carried out.



Thus, there is the potential for the amount of C in tropical soils to change over only a few years, with
potentially important consequences for atmospheric $CO_2$ concentrations.
Atmospheric $CO_2$ concentrations, and temperature, have been steadily increasing for
decades, one of the effects of this could be widespread increases in forest growth (Nemani et al.
2003) and as a result increased litterfall. Few experimental studies of the effects of elevated $CO_2$ on
forest growth have been done; Korner (2006) reported that elevated $CO_2$ caused increased litterfall
in one of three studies in steady-state tree stands in temperate forests; there have been no such
studies in the tropics. Thus the potential exists for increased $CO_2$ to increase forest growth and
litterfall – though we do not know how widespread and how large any increase in litterfall might be,
especially in the tropics.
Soil C has been shown to respond to experimental changes in litter inputs. In three studies in
temperate forests in the USA, litter removal always resulted in lower soil organic carbon, but litter
addition had much more variable effects, increasing in one (Lajtha et al. 2014a), not changing in the
second (Bowden et al. 2014) and decreasing in the third (Lajtha et al. 2014b). The single study from
the tropics, in lowland rain forest in Southwestern Costa Rica, reported decreased soil C in LR and
increased soil C in LA (Leff et al. 2012). It is therefore likely that many, but not all, forests will show
increased C in soils as a result of increased litter input.
The relative importance of aboveground or below ground inputs as sources of soil organic
matter has been reassessed in the last decade (Schmidt et al. 2011). Recently it was shown that 50-
70 % of the soil organic matter in boreal coniferous forest is from roots and root associated micro-
organisms (Clemmensen et al. 2013). The origin of the soil organic matter is thus a question of the
relative contributions of above-ground and below-ground inputs, how much of this is from microbes.
Litter manipulation experiments can provide insights into this issue by controlling one source of C
input – aboveground litterfall.
Soil nutrients as well as C can change as a result of increasing or decreasing litter inputs and
are important because they will potentially affect soil fertility. In Panama, mineralization of organic P
in only the top 2 cm of soil following three years of litter removal was calculated to be sufficient to
supply 20% of the P needed to sustain forest growth – there were corresponding increases in organic
P in litter addition plots; total nitrogen (N) showed a similar pattern (Vincent et al. 2010). 'Available'
nutrients, including KCl-extractable ammonium ($NH_4$) and nitrate ($NO_3$), and Mehlich-III extractable
phosphorus (P), potassium (K), calcium (Ca), magnesium (Mg), and micronutrients all changed over 4
years in the upper 2 cm of soil as a result of litter manipulation (Sayer and Tanner 2010). After six
years of litter manipulation surface soils (0-10 cm) had lower $NO_3$ and K in litter removal plots, and
higher $NO_3$ and Zn in litter addition plots; other nutrients were not significantly affected (Sayer et al.
2012). In Costa Rica after 2.5 years of litter manipulation surface soils (0-10 cm) had lower
nitrification in both litter removal and addition treatments, while $NH_4$ concentrations were
significantly lower in litter removal plots ($NH_4$ was 83-91% of the extractable N; Weider et al. 2013).
Thus, several soil nutrients in surface soils have been shown to change as a result of litter
manipulation but there is no consistent pattern for N, very little data for P or cations (the latter were
not reported for the Costa Rican experiment), and no data for soils deeper than 10 cm.
Here we report results from the Gigante Litter Manipulation Plots (GLiMP) experiment over
a much greater soil depth (0–200 cm) for total C, N, and P, and extractable ('plant-available') N, P, K,
Ca, Mg, manganese (Mn), and zinc (Zn), measured after 6 years of continuous litter transfer. In
addition, we present a new way of expressing soil C (relative to the unchanging mineral mass), which
allows us to calculate overall changes in soil C and other elements independently of changes in bulk



density. Our objective was to describe changes in C and nutrient concentrations in the full soil profile
and to calculate C budgets to discover what happens to the increased C input in litter addition plots.
In particular, we aimed to calculate the proportion of the added C that remains in the soil and the
litter standing crop, and can thus be considered as partial mitigation for increased forest productivity
due to increased atmospheric $CO_2$ and temperature – mitigation because C that is not in the soil will
be in the atmosphere as extra $CO_2$. No other study has tried to quantify the fate of C in organic
matter added to tropical forest soils; though a study of agricultural soil in temperate UK calculated
that about 2.4% of organic matter in yearly-added farmyard manure was still in the soil after 120
years (Powlson et al. 2011).

## 2  Materials and methods

"The study was carried out as part of an ongoing long-term litter manipulation experiment to
investigate the importance of litterfall in the C dynamics and nutrient cycling of tropical forests. The
forest under study is an old-growth semi-evergreen lowland tropical forest, located on the Gigante
Peninsula (9°06´N, 79°54´W) of the Barro Colorado Nature Monument in Panama, Central America.
The soil is an Oxisol with a pH of 4.5–5.0, with low 'available' P concentration, but high base
saturation and cation exchange capacity. Nearby Barro Colorado Island (c. 5 km from the study site)
receives a mean annual rainfall of 2600 mm and has an average temperature of 27°C. There is a
strong dry season from January to April with a median rainfall of less than 100 mm per month;
almost 90 % of the annual precipitation occurs during the rainy season. Fifteen 45-m x 45-m plots
were established within a 40-ha area (500 x 680-m) of old growth forest in 2000. In 2001 all 15 plots
were trenched to a depth of 0.5 m in order to minimize lateral nutrient- and water movement via
the root/mycorrhizal network; the trenches were double-lined with plastic and backfilled. Starting in
January 2003, the litter (including branches <20 mm in diameter) in five plots was raked up once a
month, resulting in low, but not entirely absent, litter standing crop (L- plots). The removed litter
was immediately spread on five further plots (L+ plots); five plots were left as controls (CT plots). The
assignment of treatments was made on a stratified random basis, stratified by total litterfall per plot
in 2002, i.e. the three plots with highest litterfall were randomly assigned to treatments, then the
next three and so on." (Sayer et al. 2007).  The plots were geographically blocked, litter from a
particular LR plot was always added to a particular LA plot and there was a nearby control plot.

115           Soils samples were collected in January 2009, the early dry season, using a 7.6 cm diameter
corer for the top 20 cm of soil and 2.5 cm diameter auger from 20 – 200 cm. Soil mineral
concentrations. Fresh-soil extracts for mineral nutrients were prepared within 24 h of collection
(except for $NO_3$ and $NH_4$, which were extracted within 2 hours of sampling in a 2 M KCl solution) and
determined by automated colorimetry; soil P and cations were determined by Mehlich III extraction
and analyzed by ICP-OES, soil pH was measured on a 1:2 fresh soil solution in distilled water. Dried
and ground soil was analyzed for total C and N by combustion and gas chromatography on a Flash
1112 analyzer (Thermo, Bremen, Germany). Total P was determined by ignition at 550°C for 1 h and
extraction for 16 h in 1 M H2SO4, with detection by automated molybdate colorimetry at 880 nm
using a Lachat Quikchem 8500 (Hach Ltd, Loveland, CO).

125           Nutrient data was analysed using mixed effects models, with 'litter treatment', 'depth', and
their interaction as fixed effects, and 'plot' as a random effect. Where nutrient concentrations varied
non-linearly with depth, we used splines with two or three knots. Some nutrients showed severe
heteroscedascicity, and we accounted for this in the model by using 'variance covariates', which
model the variance as a function of one or more of the effects in the model (Pinheiro and Bates
2000; Zuur et al. 2009). For all nutrients, depth was modelled as a numeric predictor and log
transformed prior to analysis. We performed model selection based on likelihood ratio tests and





Aikake Information Criterion with correction for small sample sizes (AICc, Burnham and Anderson
2002). We derived P-values for fixed effects by comparing null models to full models using likelihood
ratio tests. Final models were refitted using restricted maximum likelihood estimation (REML) (Zuur
2009). Where the treatment * depth term was significant, we refitted the model omitting either the
litter addition treatment or the litter removal treatment to assess the contribution of each of the
treatments (litter addition and litter removal) to the overall interaction term.  Analyses were done in
R version 3.1.2.

139         Soil total carbon and total nitrogen amounts were also calculated relative to soil mineral
mass to allow comparisons between the treatments where bulk density and soil depth was changing
due to removal and addition of litter; soil in litter removal plots was shrinking and had increasing
bulk density, soil in litter addition plots was increasing in depth and had lower bulk density.
Expressing potentially changing elements relative to unchanging mineral mass allows for change to
be expressed against an unchanging reference; it is analogous to expressing soil water relative to soil
dry mass rather than soil fresh mass. Soil organic C with depth was calculated for each plot by fitting
a line to cumulative soil organic C (Y) against cumulative soil mineral mass (X). Bulk density data
were measured for each plot only in the top 0-5 cm for soil. Below that we used bulk density data for
one pit only. Bulk density below 10 cm depth does not vary much across the site; data for four soil
pits (not in any of the plots) have a coefficient of variation of about 10 % for soils from 10 - 20 cm
deep and 3 % for soils from 20-50 cm deep), whereas coefficients of variation of bulk densities in
surface 0-5 cm soils were higher: control 12 %, LA 15 % and LR 4.9 %. Bulk density data were used to
estimate approximate soil depth for control plots in Fig. 4. Statistical comparisons of modelled
cumulative total C against cumulative mineral matter were compared by bootstrapping, using R
version 3.1.2.

## 3  Results

Soils in LA plots, compared to LR plots, had significantly higher: $NO_3$ and pH (to 30 cm); $P_{Meh}$ and total
C (both to 20 cm); total N (to 15 cm); Ca (to 10 cm); Mg and lower bulk density (both to 5 cm), (Figs 1
and 2 and Table S1). When compared to control soils, there were fewer differences, LA soils had
higher concentrations of $P_{Meh}$ (to 20 cm); $NO_3$ (to 15 cm); Ca (to 10 cm); and pH (to 10 cm). LR soil
nutrient concentrations were not significantly lower than those in controls. Nutrient concentrations
in soils > 30 cm deep did not differ significantly for any nutrient. Thus, in some way total C, total N,
$NO_3$, $P_{Meh}$, Ca and Mg  were significantly affected by litter removal or addition, but K, Mn, $NH_4$, Zn
and were not; effect sizes (log response ratio for 0-5 cm soils) decreased from  0.81 for $NO_3$, to 0.39
for Ca, 0.27 for Zn, 0.20 for $P_{Meh}$, 0.20 for Mg, 0.15 for $C_{tot}$, 0.11 for $N_{tot}$.

166         All nutrients decreased in concentration with increasing soil depth. In control soils,
concentrations at 50–100 cm compared to 0–5 cm were: $NH_4$ 50 %, Mg 37 %, $P_{tot}$ 36 %,  K 32 %, $P_{Meh}$
25 %, $NO_3$ 24 %, $N_{tot}$ 12 %, Ca 11 % and $C_{tot}$ 11 %;  $NO_3$ was only 24 % of the total inorganic N in
controls (mean over all depths) (Figs 1 and 2 and Table S1). Concentrations of most elements
continued to decrease below 100 cm deep in the soil; those from 150–200 cm were about half those
from 50–100 (ranging from 14% for Ca to 81% for $NH_4$, Table S1).

172         Soil bulk density in the top 5 cm was significantly lower in LA than LR, though neither was
significantly different from the controls. Soil C stocks standardized to a consistent mineral mass (*i.e.*
that in the control plots) was significantly greater in LA compared to LR to about 10 cm deep in the
soil (Figs 3 and 4). Total N per mineral mass of soil was also significantly greater in LA than LR in





approximately the top 10 cm of soil. In contrast, C:N ratios changed little with depth; in control soils,
C:N was about 10.5 near the surface and 10.0 at 150–200 cm, in LR plots, C:N was 10.5 at the surface
and 10.3 at depth, while LA soils were more variable, with C:N being 11.7 at the surface and about
10.0 at 150–200 cm deep.

**4   Discussion**
**4.1 Soil carbon dynamics**
The amount of C 'missing' from LR and 'extra' in the LA over about the top 20 cm of soil
(from calculations based on C per mineral matter), six years after litter removal and addition started
(January 2009), was about 0.5 kg C m$^{-2}$ (Fig. 3). The similarity of the losses from LR and gains in LA
probably has different causes: we speculate that losses from the soil in the LR plots are due to
respiration being greater than additions; we did not physically remove organic matter from the
mineral soil. We further speculate that increases in C in the mineral soil in the LA plots are a result of
infiltration of dissolved and suspended organic matter draining from the litter standing crop, and/or
changes in root exudates; increases in root growth are not the explanation – root growth was lower
in LA plots (Sayer et al. 2006).
In addition to the extra *soil* C in the LA plots the litter standing crop (LSC) was also higher in
LA plots; in September 2005 (2.8 years after litter manipulation started) there was 0.4 kg C m$^{-2}$ extra
in the Oi and Oe layers compared to control plots (Sayer and Tanner 2010) and data from 2013 show
that LSC was at about this level (C. Rodtassana in prep.). Together this extra 0.9 kg C m$^{-2}$ in the LA
soil and litter standing crop is about 30 % of the 3 kg C m$^{-2}$ in litter added to the LA plots over 6 years
(litterfall is c.1 kg m$^{-2}$ yr$^{-1}$, c. 45 % is C, times 6 years). This increase in C in surface soil and the litter
standing crop could be interpreted as *potential* partial mitigation of the effects of increasing $CO_2$
concentrations in the atmosphere, though any increases in litterfall due to increased $CO_2$ will be less
than our experimental doubling (a free air CO2 experiment in 13-year old loblolly pine plantation in
North Carolina U.S.A reported a 12% increase in litterfall over 9 years (Lichter et al 2005 and 2008)).
The increases in soil C in our LA plots (c. 1% per year, of total C to c. 20 cm depth) are much
smaller than those reported in the other study of litter manipulation in tropical forest (lowland rain
forest in Southwestern Costa Rica) where two years of removing litter reduced soil C concentration,
in the top 10 cm of soil, by 26 % and doubling litter increased soil C by 31 % (Leff et al. 2012). In
three temperate forest studies rates of change in soil C were low; but they were measured over
much longer periods. In north central USA soil C content decreased by 61 % in litter removal plots
and increased by 33 % in double litter plots over a 50-year period (Lajtha et al. 2014a). In
Pennsylvania, USA, 20 years of removing litter reduced soil C by 24%, although the corresponding
litter doubling had no effect (Bowden et al. 2014). In deciduous forest in Massachusetts, USA, 20
years of LR also reduced mineral soil C (by 19%), but LA also resulted in lower mineral soil C (by 6%,
Lajtha et al. 2014b). Differences between forests in the effect of litter addition on soil organic matter
could be partly due to differences in priming of pre-existing soil organic C resulting in no, or small,
increases in soil C in double litter plots. Priming might be greater in N limited temperate forests
remote from atmospheric N pollution, because one cause of priming is mining of soil organic matter
for N by microbes stimulated by additions of litter with low N concentrations (relative to soil organic
matter) (e.g. Nottingham et al. 2015). It is therefore likely that many, but not all, forests will show
increased C in soils as a result of increased litter input.





Soil C may on average be composed more of C from roots than shoots (Rasse et al. 2005)
and that may be the case in our soils in Panama because although changes in litter inputs have
caused changes in soil C they are very small, c. 1% of total soil C per year, compared to the 'normal'
turnover of C of 25% (0-10 cm soil) within 6 months - as calculated from changes in C concentration
from wet season to dry season (Turner et al. 2015), and an annual turnover of about 7% based on
incorporation of $^{13}$C into soils over decades (Schwendenmann and Pendall 2008). Other tropical
forest soils also had high turnover rates of C; in Eastern Brazil 40-50 % of the C in the top 40 cm of
soil had been fixed in about 32 years (Trumbore 2000). In Panama the much higher rates of turnover
of soil C as compared to changes caused by litter removal and addition suggest that the main source
of soil organic matter (over months to a few years) is roots, root exudates and mycorrhizal fungi.
Nevertheless, changes in above ground litter input are still important, because they have resulted in
overall decreases and increases in soil C.

**4.2 Litter manipulation - depth of effects.**

Effects of litter removal and addition differed among nutrients and were strongest near the soil
surface, with no significant differences below 30 cm. The strength of the effects and the depth to
which they were significant are increasing with time. Four years after the start of litter manipulation
six nutrients showed significant effects in the upper 2 cm of soil (NO$_3$, NH$_4$, P$_{Meh}$, K, Ca, Mg), whereas
only NO$_3$ and Ca showed significant effects from 0-10 cm (Sayer et. al 2010). After 6 years, in the
early dry season 2009 (current paper), effects were seen to greater depths: NO$_3$ was higher to 30 cm
and P$_{meh}$, to 20 cm in LA plots. Over time significant differences have become apparent for more
nutrients and to greater depth in the soil; these differences were caused by differences in litter
input.
The concentrations of NH$_4$ and NO$_3$ are usually only measured in surface soils in tropical rain
forests, perhaps because N is generally thought not to limit growth in such forests; though
fertilization with N and K together increased growth of saplings and seedlings in the Gigante
Fertilization Project (GFP), which was adjacent to our litter manipulation experiment in Panama
(Wright et al. 2011). Relevant concentrations of NH$_4$ and NO$_3$ are also difficult to measure since they
change rapidly over only a few hours (Turner and Romero 2009); extractions for the current paper
were done within two hours of collecting soils. In our litter manipulation experiment NH$_4$ accounted
for 76% of the sum of NH$_4$ and NO$_3$ (mean over all depths in controls plots) and decreased less with
depth than NO$_3$ (at 50-100 cm NH$_4$ was about 50 % of surface values whereas N0$_3$ was about 25 %).
In the GFP Koehler et al. (2012) reported that NH$_4$ also deceased less with depth (at 200 cm it was 41
% of surface soils) than NO$_3$ (to 17 % of surface soils), and that NH$_4$ was the dominant form of total
inorganic N (about 80 %) – the same patterns as in our litter manipulation experiment. Nitrogen
dynamics in soils have also been measured in a litter manipulation experiment in Costa Rica (Wieder
et al. 2013), where nitrification rates were lower in both LR and LA plots and extractable NH$_4$ was
significantly lower in LR plots. This contrasts with our results of greater NO$_3$ in LA compared to LR
and no effect on NH$_4$; the differences between the experiments may be partly due to somewhat
different soils and a wetter climate in Costa Rica (c. 5 m rain per year c.f. 2.6 in Panama). Thus, soil N
dynamics differ somewhat between the only two tropical litter manipulation experiments, but in
both NH$_4$ was the dominant form of inorganic N, and in both total inorganic N decreased in LR plots
and increased in LA plots (though differences were not always statistically significant).
The 'available' forms of P are also not often reported for the deeper horizons of tropical
forest soils, despite the fact that P is usually regarded as the most likely limiting nutrient in such



forests (Tanner et al. 1998 and Cleveland et al. 2011) and has been shown to limit fine litter
production in the adjacent Gigante Fertilizer experiment (Wright et al. 2011). Mehlich P and total P
both decreased with depth in control soils in our litter manipulation experiment (to 25 and 29 % of
near surface values); in LR soils the decrease was less steep (37 % and 36 %). LA increased Mehlich P
in the surface soils (though total P was not higher), indicating increased P availability, which is
consistent with the finding that LA decreased the strength of phosphate sorption in these soils
(Schreeg at al. 2013). Thus for P, potentially the most commonly limiting nutrient in tropical rain
forest soils, six-years of continuous removal and addition of litter in our experiment has reduced and
increased 'available' P down to 20 cm in the soil.
The relative amounts of exchangeable cations and their change with depth in the control
plots of the Panamanian litter manipulation soils are similar to patterns in other tropical forest soils.
In our experiment, Ca concentrations (in centimoles of charge) are about twice those of Mg in
surface soils (though below 30 cm Mg to Ca ratios exceed 1); K concentrations are usually less than 5
% of the sum of exchangeable Ca, Mg and K. With increasing depth Ca, Mg and K concentrations all
decrease, with Ca decreasing more than Mg or K. Other tropical forest soils are similar; in 19 profiles
throughout Amazonia the sum of base cations (Ca, Mg, K) was usually dominated by exchangeable
Ca (11 cases) or Ca was equal to Mg (4 cases), and both Ca and Mg mostly decreased with depth,
while K was in low or in trace concentrations in all profiles (Quesada et al. 2011). In Hawaii (Porder
and Chadwick 2009), much younger soils (11,000 BP on lava), with much higher concentrations of
Ca, Mg and K than Panama and Amazonia, showed similar patterns: Ca was the dominant cation, K
was usually less than 5 % of the sum of exchangeable Ca, Mg and K, and all cations decreased with
depth at the wetter sites (but not in the drier sites). Thus in most wet tropical forest soils, Ca is the
most abundant cation and most cations decrease with depth. Litter addition in Panama increased Ca
and Mg concentrations in the surface soils and thus steepened the depth gradient, whereas litter
removal decreased Ca and Mg and therefore decreased the gradient; K was at much lower
concentrations (as in Amazonia and Hawaii) and was not affected by LA and LR even in 0-5 cm soils.
**4.3 Design of litter manipulation experiments**
The design of litter manipulation experiments needs to be carefully considered when
evaluating their results; the strength of the effect of litter manipulation on soil C in Panama was
much less than that in Costa Rica. The Panamanian and Costa Rican experiments are very different in
spatial scale. Plots in Panama are large, 45 x 45 m, those in Costa Rica are small, 3 x 3 m. The small
plots are 'hot' and 'cold' spots relative to large individual tree crown areas (and likely tree root
areas); crowns of the largest trees in lowland rain forests are commonly 25 m in diameter, so a 3 x 3
m plot is 2 % of that area. These differences in experimental design and their effects on the pattern
of the results should be considered when trying to understand ecosystem level processes; small hot
and cold spots may not represent what would happen in plots on the scale of the large trees.

**5  Conclusions**
The increase in C in the mineral soil and the litter standing crop following litter addition was
statistically significant in the top 20 cm of the soil, suggesting that any increased litterfall as a result
of increased atmospheric $CO_2$ and/or temperature could result in a substantial increase in soil C and
therefore partially mitigate the increase in atmospheric $CO_2$. However, the current experiment
added much more litter than might be produced by an increase in $CO_2$ of, say, 200 ppm, and added
more nutrients than might occur even in polluted sites. Thus new experiments are required to




investigate the effects of more realistic increases in litterfall using litter with low nutrient
concentrations.
Supplementary material
Table S1 with full original data from soil analyses
Table S2 Model estimates of concentrations (from Sheldrake)
*Acknowledgements*. We thank J. Bee, L. Hayes, S. Queenborough, R. Upson and M. Vorontsova for
surveying the plots, J Bee for setting up the experiment in 2000 and 2001; E. Sayer for running the
experiment from 2001-2009; A Vincent for helping to maintain the experiment from 2003-2005. T.
Jucker did the statistics to compare the effect of treatment on soil C relative to mineral matter.
Funding for the project was originally from the Mellon Foundation (1999-2002); on-going costs were
paid for by the Gates-Cambridge Trust (E Sayer); The University of Cambridge Domestic Research
Studentship Scheme and the Wolfson College Alice Evans Fund (A. Vincent); The Drummond Fund of
Gonville and Caius College and Cambridge University (E. Tanner). The whole of the experiment
depended on the continuous raking of litter; which was done by Jesus Valdez and Francisco Valdez.
S.J. Wright has been a frequent source of help for many aspects of the experiment.

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

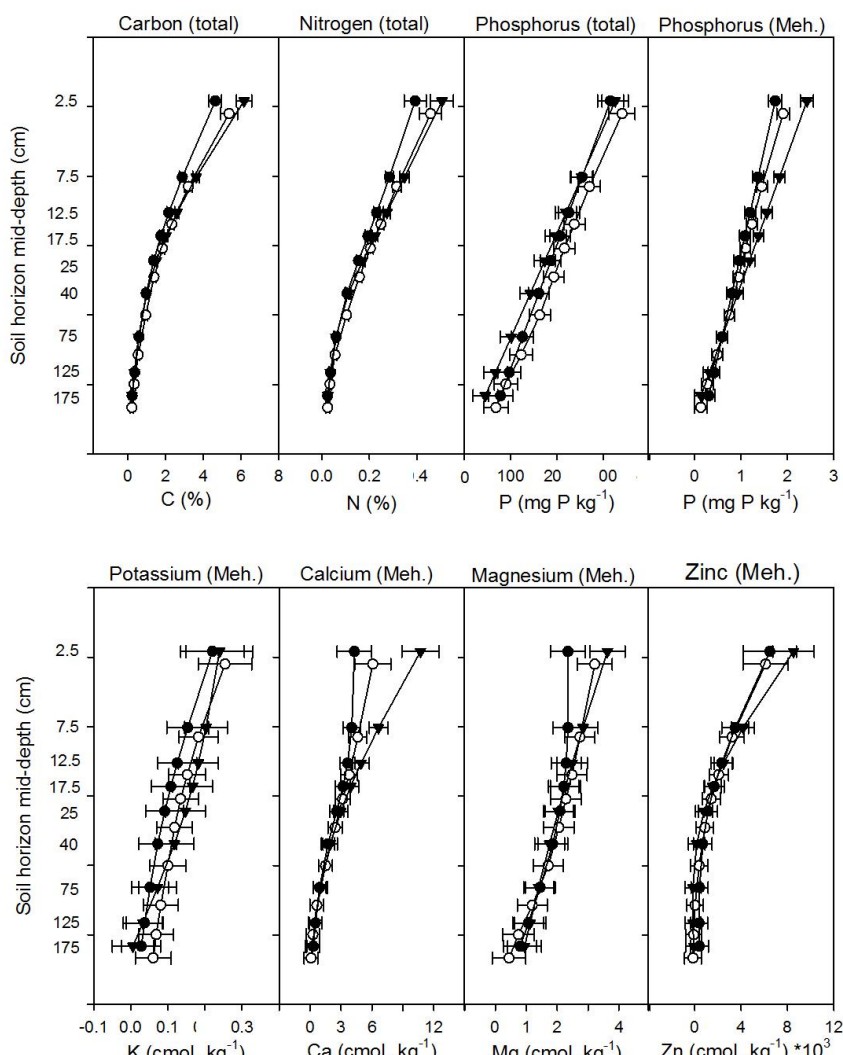


Fig. 1 Concentrations of soil C, N, P (various fractions) and cations (Mehlich extractions), plotted
against the mid-point of the soil layers sampled (Zn values should be divided by 1000 to obtain
actual means), control points are displaced below treatments. Data are fitted values of the mixed
effects models with 95% confidence intervals (see Methods), in litter removal ●, control O and litter
addition ▼ plots.







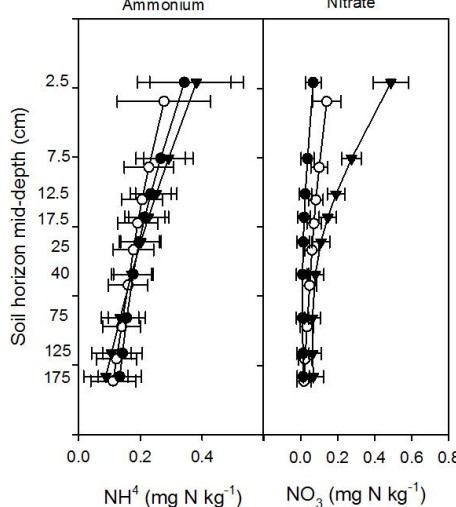


Fig. 2 Mean concentrations of ammonium and nitrate plotted against the mid-point of the soil layers
sampled, control points are displaced below treatments. Data are fitted values of the mixed effects
models with 95% confidence intervals (see Methods), in litter removal ●, control O and litter
addition ▼ plots.




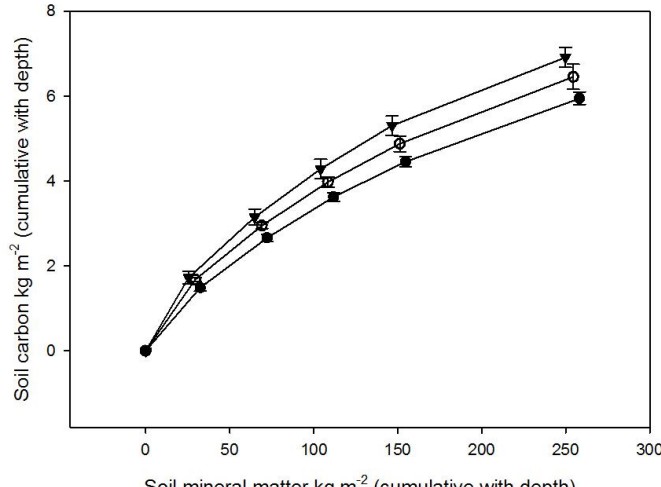



Fig. 3 Soil carbon content and mineral content in litter addition, control, and litter addition
expressed as kg C m⁻² cumulatively from 0 to 30 cm soil depth. Values are means for 5 plots per
treatment +/- SE, litter removal ●, control O, and litter addition ▼ .





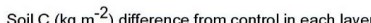

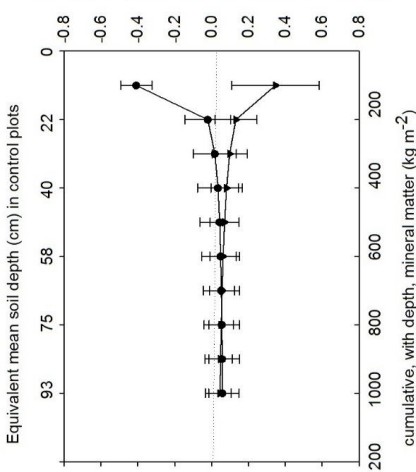


Fig. 4 Differences in soil carbon content relative to control soils (mean and SE, n = 5), after 6 years of
litter manipulation, plotted for successive soil layers: 0-100 kg (mineral matter) m$^{-2}$, plotted at 100 kg
m$^{-2}$ on right y axis; 100-200 kg m$^{-2}$, plotted at 200 kg m$^{-2}$; and so on to 900-1000 kg m$^{-2}$, plotted at
1000 kg m$^{-2}$; in litter removal ● and litter addition ▼ plots. We calculated the soil C in the LR and
LA plots at the mineral mass equal to that at various depths in the control plots (0-5 cm, 5-10 cm,
etc), we then calculated the difference in C between each litter removal (or litter addition) and its
control plot for the same mineral mass. Approximate depth for cumulative soil mineral mass in
control plots is shown on left y axis.

451