# Peer review of "Changes in soil carbon and nutrients following six years of litter removal and addition in a tropical"

_Biogeosciences, 2016_

## Referee Comment (RC1) · W. Wieder (Referee) · 26 Aug 2016

General comments

Tanner and co-authors present an interesting analysis on the changes in soil chemistry following a large-scale litter manipulation in a tropical rain forest. They nicely contextualize their results with findings from other studies while clearly and concisely describing their findings. The manuscript makes an important contribution to our understanding of potential soil biogeochemical response to changes in plant litterfall, and with minor changes and clarifications the manuscript should be acceptable for publication in Biogeosciences.

[Figure]

Specific comments

It may be appropriate to express changes in soil C stocks as a function of soil mineral mass, but a description of how this was measured is missing from the text. Was soil mineral mass measured in each plot, in each treatment, or in a single pit (like bulk density)?

More broadly, the emphasis placed on soil mineral mass to extrapolate findings seems somewhat surprising, and I'd suspect it's driven by either a lack of appropriate bulk density data for scaling, or non-significant results when using the available data. Either way I'm not asking the authors to go out and take more measurements, but would appreciate greater transparency to understand their decision to focus on soil mineral mass. If data are available to make an extrapolation of Fig 3 with depth on the X axis it would be much more valuable for studies trying to quantify or model changes in soil C stocks, as information about mineral mass is typically lacking or not considered.

I recall publications from some of the temperate DIRT plots (e.g., Lajtha references in the paper) showed changes in different soil C fractions. I assume similar data are not available for this study, but I wonder if consideration of C stabilization mechanisms and soil mineralogical conditions could help explain some of the differences between temperate and tropical sites. Is it worth a brief discussion on this point (e.g. expanding / developing the paragraph that begins on line 202)?

The authors (justifiably) seem keen on their soil P results, which are interesting and relevant (line 262). Is it possible to extrapolate findings for P, similar to the soil C figure 3, making this a multi-panel figure? I think this would illustrate the conclusion that experimental manipulations modify soil nutrient cycles in (perhaps) unexpected ways.

The discussion starts of with the introduction of new results. I appreciate the authors wanting to focus readers' attention on these findings, but feel like results (Figs 3 & 4) are best introduced in the results, not discussion section of a paper.

[Figure]

**BGD**

Finally, calling out the small plots from the Costa Rican study seems a bit unjustified in a single paragraph subsection of the discussion. Granted the authors make a good point about the appropriate size of experimental plots, but I think Leff and co-authors (2012, cited in the paper) acknowledge the limitation of their small plots. If the authors want this section to remain they should more broadly discuss other litter manipulation studies, not just the Costa Rican site.

Technical corrections:

Introduction: specific values for C pools, turnover times, and fractions seem unnecessarily detailed (lines 33, 36). More broadly the introductions reads a bit like a bullet point of disconnected ideas. This is a stylistic concern, not a scientific one.

Throughout, check that abbreviations are defined before they are used in the text (eg. LR and LA line 55, GFP line 251).

Line 66-68, This is unclear P mineralization (0-2 cm) in LR plots met 20% of NPP needs, or the decline in P mineralization would have met this demand?

Line 76. This study looked at net nitrification and should be Wieder et al. 2013 (i before e).

Line 89. Awkward. Forest productivity isn't mitigated, but increases in terrestrial C storage can mitigate atmospheric CO2 accumulation. Line 210. Awkward, maybe insert 'a' here: In a deciduous forest in MA. . .

Line 307. What is meant by 'polluted' sites? Is this sites receiving large amounts of N, P or micronutrient deposition (is the later actually a real a thing)? Is this just to say that litter manipulations aren't identical to CO2 enrichment alone, because they also serve as nutrient manipulations that modify ecosystem dynamics?

---

## Referee Comment (RC2) · Anonymous Referee #2 · 3 Oct 2016

This manuscript nicely describes the effect of litter manipulation on soil carbon and nutrient stocks in tropical rainforest. I appreciated the efforts the authors made to improve the manuscript based on previous reviewer suggestions. In particular, revisions made to the Introduction and Discussion sections have made this manuscript much easier to read.

I agree with many of the already published referee comments (bg-2016-229-RC1) that there are still a few clarifications that could be made to improve the manuscript prior to final publication. While most of my comments were technical, a few were more substantive.

General comment: Referee comment 1 points out that a few more details about how

soil mineral mass was measured would be valuable, and I concur. Likewise, I would appreciate seeing a comparison of results using more traditional ways of measuring soil C (e.g., fraction of dry mass) and the approach utilized here. Given its novelty, mineral mass is of limited utility when comparing to other studies. If there were no significant differences in nutrient content among manipulations or depths using other approaches, this would speak to the importance of using this method to calculate ecologically meaningful change in soil C.

Technical comments: Please clarify abbreviations: The LA and LR abbreviations were not spelled out when the appeared for the first time on lines 55 and 56, and appear to have been conflated with L- and L+ in the Sayer et al. quote on lines 109 and 110.

The sentence that begins on line 75 is awkward - perhaps a better way of saying this is that "After 2.5 years of litter manipulation in Costa Rica, surface soils (0-10 cm) had lower nitrification in both litter removal and addition treatments..."

On line 89, the carbon that stays in soil and litter crop does not mitigate increased forest productivity - it mitigates increased atmospheric C, or something like that. This line was confusing and, as stated, does not appear to be correct.

I appreciated the improvements to the figures in response to previous comments. The figures could be strengthened by including notations to depict which litter effects were significantly different from controls. While this information is largely contained in the text, including this in the figures would help if the images were ever reproduced for other uses.

---

## Author Comment (AC1) · 19 Oct 2016

Tanner responses to comments by W Wieder.

1) **Comment.**

"Was soil mineral mass measured in each plot, in each treatment, or in a single pit (like bulk density)"

Response

Soil mineral matter was calculated for each plot and soil depth from soil carbon concentration (mineral matter is total soil mass minus twice soil carbon content). Bulk density was measured for every plot for 0-5 cm depth; below 5 cm we used the bulk density from one soil pit (lines 146-151 in the manuscript).

2) **Comment.**

"More broadly, the emphasis placed on soil mineral mass to extrapolate findings seems somewhat surprising,"

Response

The emphasis on expressing soil carbon per mineral mass is to deal with the (general) problem that as soil organic matter changes the bulk density changes, so sampling to the same depth will not be comparing like with like. This is well known problem - Powlson et al 2011 say "The principle is that an equal mass of organic-matter-free mineral soil should be sampled between the treatments or times being compared." For this reason in our study in Panama we expressed carbon relative to an unchanging mineral mass. It is also an easy calculation to make and can often be made retrospectively on published data. It was not done to get round a problem of non-significant results.

3) **Comment.**

"If data are available to make an extrapolation of Fig 3 with depth on the X axis it would be much more valuable for studies trying to quantify or model changes in soil C stocks, as information about mineral mass is typically lacking or not considered."

Response

Fig 4 shows the cumulative (with depth) mineral matter and soil depth in the control plots, down to about 93 cm. An e mail exchange with the referee clarified that he wanted a second axis In Fig 3 showing the soil depth in the control plots – we have done this. We disagree with the comment that "information about mineral matter is typically lacking", because if samples have data on soil carbon per dry soil mass, then the mineral matter is easily calculated (as total mass minus twice soil carbon - there will be a small error because soil organic matter is not exactly twice soil carbon, but the effect will be trivial.)

4) **Comment.**

"I recall publications from some of the temperate DIRT plots (e.g., Lajtha references in the paper) showed changes in different soil C fractions. I assume similar data are not available for this study, but I wonder if consideration of C stabilization mechanisms and soil mineralogical conditions could

help explain some of the differences between temperate and tropical sites. Is it worth a brief discussion on this point (e.g. expanding / developing the paragraph that begins on line 202)?"

Response

Other researchers are working on this in the experiment. As we present no data on carbon fractions in this paper we think it best to leave discussion of that subsequent manuscripts.

5) **Comment.**

"The authors (justifiably) seem keen on their soil P results, which are interesting and relevant (line 262). Is it possible to extrapolate findings for P, similar to the soil C figure 3, making this a multi-panel figure?"

Response

It is not sensible to express cumulative Mehlich P per cumulative mineral mater (in an analogous way to cumulative carbon per cumulative mineral matter in Fig. 3) because a substantial (but unknown) amount of Mehlich P comes from organic matter. Soil matter is either organic or mineral and we plot one against the other in Fig. 3; Mehlich P is different - it comes from both mineral and organic matter.

6) **Comment.**

"The discussion starts off with the introduction of new results. I appreciate the authors wanting to focus readers' attention on these findings, but feel like results (Figs 3 & 4) are best introduced in the results, not discussion section of a paper"

Response

We disagree. The 'results' are concentrations of carbon per mass. We then use those results to calculate concentrations of carbon per mineral matter.

7) **Comment.**

"Finally, calling out the small plots from the Costa Rican study seems a bit unjustified in a single paragraph subsection of the discussion. Granted the authors make a good point about the appropriate size of experimental plots, but I think Leff and co-authors (2012, cited in the paper) acknowledge the limitation of their small plots. If the authors want this section to remain they should more broadly discuss other litter manipulation studies, not just the Costa Rican site."

Response

We are not making any personal points here, but we do think that there is a real issue about the size of experimental plots affecting the qualitative patterns of results. Specifically, small (3 x 3 m) litter removal and addition plots might be local cold spots and hot spots that will affect the responses. The pattern of results from small plots might be the OPPOSITE of those from large plots. For example, small litter addition plots might cause extra root growth into local patches of soil with extra nutrients, but large litter addition plots (45 x 45 m) might cause reduced root growth because the whole tree is receiving extra nutrients and 'can afford' to reduce root growth and put more into shoot growth, in other words, a completely opposite pattern of results caused by differences in experimental design. We simply want to point out that the design of these experiments might well affect the pattern of results. If there were lots of experiments like this we could look for patterns, but there aren't many.

To address the reviewer's comment, we have changed the last line to "small hot and cold spots may not represent what would happen in plots on the scale of the large trees - as pointed out by Leff et al 2012."

8) Technical corrections:
Comment. Introduction: specific values for C pools, turnover times, and fractions seem unnecessarily detailed (lines 33, 36). More broadly the introductions reads a bit like a bullet point of disconnected ideas. This is a stylistic concern, not a scientific one.

Response. As Wieder says this is stylist – we think this is clear and informative

**Comment.** Throughout, check that abbreviations are defined before they are used in the text (eg. LR and LA line 55, GFP line 251).

Response. We have changed all 'LR' to 'litter removal' and all 'LA' to 'litter addition'. We have reworded the text so that GFP is no longer used.

**Comment.** Line 66-68, This is unclear P mineralization (0-2 cm) in LR plots met 20% of NPP needs, or the decline in P mineralization would have met this demand?

Changed to "mineralization of organic phosphorus (P) (inferred from the decrease in the concentration of organic P) in the top 2 cm of soil during three years of litter removal was calculated to be sufficient to supply 20% of the P needed to sustain forest growth"

**Comment.** Line 76. This study looked at net nitrification and should be Wieder et al. 2013 (i before e).

Response. Added 'net' and corrected spelling of Wieder.

**Comment.** Line 89. Awkward. Forest productivity isn't mitigated, but increases in terrestrial C storage can mitigate atmospheric CO2 accumulation.

Response. Changed to "can thus be considered as partial mitigation of atmospheric $CO_2$ accumulation through increased forest productivity"

**Comment.** Line 210. Awkward, maybe insert 'a' here: In a deciduous forest in MA. . .

Response. Changed to "In a deciduous forest in"

**Comment.** Line 307. What is meant by 'polluted' sites? Is this sites receiving large amounts of N, P or micronutrient deposition (is the later actually a real a thing)? Is this just to say that litter manipulations aren't identical to CO2 enrichment alone, because they also serve as nutrient manipulations that modify ecosystem dynamics?

Response. This site is not receiving large amounts of N or P (though N input is increasing Hietz et al 2011 Science 334, 664). Our comparisons are based on N & P inputs in polluted sites in USA and Europe. We have added 'temperate'.

We don't mention micronutrients in the Conclusions – so we ignore that part of the comment.

---

## Author Comment (AC2) · 19 Oct 2016

Biogeosciences Discuss. of Tanner at al. "**Changes in soil carbon and nutrients**"

Author responses to reviewer 2

Comment "I would appreciate seeing a comparison of results using more traditional ways of measuring soil C (e.g., fraction of dry mass) and the approach utilized here. Given its novelty, mineral mass is of limited utility when comparing to other studies."

Response

Tanner did a quick calculation (using the data in the supplementary material) of the changes in concentration over the top 20 cm of soil. Litter removal soil shows a 1.9% fall in concentration and litter addition a 2.0% increase in C concentration. This compares with 1% per year using the 'new' calculation based on the same amount of mineral matter. We could put a sentence about this into a revised ms. E.g. "The increases in soil C in our litter addition plots (c. 1% per year, of total C to c. 20 cm depth)"; this is about half of the change calculated using fixed depths and % carbon concentrations (2% per year). "Our changes are much smaller.."

Comment

Technical comments: Please clarify abbreviations: The LA and LR

Response. LA and LR now written out in full everywhere. L- and L+ now changed to litter removal and litter addition.

Comment

The sentence that begins on line 75 is awkward - perhaps a better way of saying this is that "After 2.5 years of litter manipulation in Costa Rica, surface soils (0-10 cm) had lower nitrification in both litter removal and addition treatments..."

Response

We ask to keep our original wording. We deliberately put "In Cost Rica" first in the sentence to mark the fact that we are moving on in the discussion from Panama to Costa Rica. If we start with "After 2.5 years of litter manipulation" it could be taken to mean that we are still discussing Panama.

Comment

"On line 89, the carbon that stays in soil and litter crop does not mitigate increased forest productivity"

Response.

I could not find this. Anyway, in our revised ms we use 'mitigate' only once

"The increase in C in the mineral soil and the litter standing crop following litter addition was statistically significant in the top 20 cm of the soil, suggesting that any increased litterfall as a result of increased atmospheric $CO_2$ and/or temperature could result in a substantial increase in soil C and therefore partially mitigate the increase in atmospheric $CO_2$."

Comment

I appreciated the improvements to the figures in response to previous comments. The figures could be strengthened by including notations to depict which litter effects were significantly different from controls. While this information is largely contained in the text, including this in the figures would help if the images were ever reproduced for other uses.

Response.

Win Figs 2 & 2 we plot means and confidence errors if errors don't overlap means are usually significantly different; we say which are significant in the text. Also we make comparisons between litter removal and litter addition, as well as between each treatment and control, showing both types of comparison on the figure might clutter up the diagrams, but if the Editor thinks it useful we will do it.